# Self-Adapting Language Models

**Adam Zweiger**[*] **Jyothish Pari**[*†] **Han Guo** **Yoon Kim** **Pulkit Agrawal**[†]
Massachusetts Institute of Technology
{adamz, jyop, hanguo, yoonkim, pulkitag}@mit.edu

## Abstract

Large language models (LLMs) are powerful but static; they lack mechanisms to adapt their weights in response to new tasks, knowledge, or examples. We introduce **Se**lf-**A**dapting **L**LMs (SEAL), a framework that enables LLMs to self-adapt by generating their own finetuning data and update directives. Given a new input, the model produces a *self-edit*—a generation that may restructure the information in different ways, specify optimization hyperparameters, or invoke tools for data augmentation and gradient-based updates. Through supervised finetuning (SFT), these self-edits result in persistent weight updates, enabling lasting adaptation. To train the model to produce effective self-edits, we use a reinforcement learning loop, using the downstream performance of the updated model as the reward signal. Unlike prior approaches that rely on separate adaptation modules or auxiliary networks, SEAL directly uses the model's generation to parameterize and control its own adaptation process. Experiments on knowledge incorporation and few-shot generalization show that SEAL is a promising step toward language models capable of self-directed adaptation in response to new data. Our website and code is available at `https://jyopari.github.io/posts/seal`.

## 1 Introduction

Large language models (LLMs) pretrained on vast text corpora exhibit remarkable abilities in language understanding and generation [1, 2, 3, 4, 5]. However, adapting these powerful models for specific tasks [6], integrating new information [7], or mastering novel reasoning skills [8] remains challenging due to the limited availability of task-specific data. In this paper, we explore an intriguing hypothesis: can an LLM self-adapt by transforming or generating its own training data and learning procedure?

As an analogy, consider a human student preparing for the final exam of a machine learning class. Many students rely on their notes to prepare for the exam. These notes are often derived from the lecture content, textbooks, or information available on the internet. Instead of relying on the raw content, assimilating and rewriting the information in the form of notes often improves the ability of students to understand the content and answer exam questions. This phenomenon of reinterpreting and augmenting external knowledge in a way that is easier to understand is not limited to just taking exams, but seems to be universally true of human learning across tasks. Furthermore, different humans assimilate information in different ways—some might condense the information into a visual diagram, some into text, or some might rely more on concrete mathematical descriptions.

Such assimilation, restructuring, or rewriting of data as part of the learning process is in contrast with how large language models (LLMs) are typically trained and deployed. Given a new task, current LLMs consume and learn from the task data "as-is" via finetuning or in-context learning [9, 10, 11, 12]. However, such data may not be in an optimal format (or volume) for learning, and

---

[*]Equal contribution.
[†]Improbable AI Lab, CSAIL MIT

39th Conference on Neural Information Processing Systems (NeurIPS 2025).

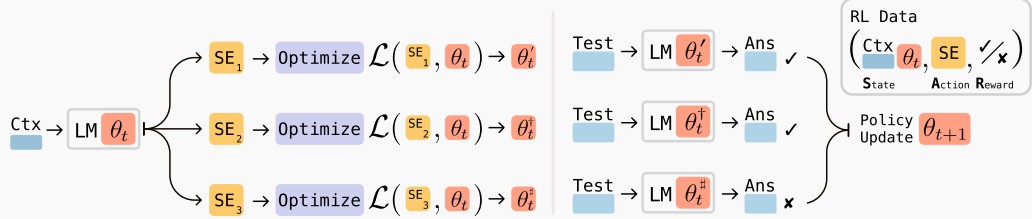

Figure 1: **Overview of SEAL.** In each RL outer loop iteration, the model generates candidate self-edits (SE)—directives on how to update the weights—applies updates, evaluates performance on a downstream task, and uses the resulting rewards to improve the self-edit generation policy.

current approaches do not enable models to develop bespoke strategies for how to best transform and learn from their training data.

As a step towards better language model adaptation, we propose equipping LLMs with the ability to generate their own training data and finetuning directives in response to new inputs. In particular, we introduce a reinforcement learning algorithm that trains LLMs to generate **"self-edits"**—natural-language instructions that specify the data and, optionally, optimization hyperparameters for updating the model's weights (see Figure 1). We refer to such models as **Se**lf-**A**dapting **L**LMs (SEAL).

We evaluate SEAL on two applications. We first consider the task of integrating new factual knowledge into an LLM. Rather than finetuning directly on the passage text, we finetune on synthetic data generated by the SEAL model. Our results show that, following reinforcement learning (RL) training, finetuning on self-generated synthetic data improves question-answering performance on the no-passage-in-context variant of SQuAD [13] from 33.5% to 47.0%. Notably, self-generated data from SEAL outperforms synthetic data generated by GPT-4.1.

We further evaluate SEAL on few-shot learning on a simplified subset of the ARC-AGI benchmark [14], where the model leverages a set of *tools* to autonomously select both synthetic data augmentations and optimization hyperparameters (e.g., learning rate, training epochs, selective loss computation over token types). Our experiments demonstrate that automatic selection and configuration of these tools using SEAL enhances performance compared to both standard in-context learning (ICL) and self-editing *without* RL training to use the tools effectively. These results collectively show that SEAL is a versatile framework for enabling language models to self-adapt.

## 2 Related Work

**Synthetic Data Generation.** The creation of synthetic data for LLM training is increasingly common, from large-scale pretraining datasets [15, 16, 17, 18, 19] to task-specific data augmentation [20, 21, 22] and instruction-tuning sets [23, 24]. For incorporation of a smaller corpus, Yang et al. [25] use synthetic data generation via graph-based prompting. SEAL builds on this line of work by using reinforcement learning to train a *generative policy* that directly maximizes the downstream utility of synthetic data when applied for gradient-based self-updates, rather than relying on static or heuristic generation strategies that are manually tuned.

**Knowledge Updating.** Several recent works aim to modify or inject factual knowledge into language models via weight updates. Some methods attempt to directly locate specific parameters that correspond to individual facts [26, 27, 28]. Others propose generating additional finetuning data using the information in context [29, 30, 25, 31, 32]. We adopt the latter strategy, following Akyürek et al. [30], who propose generating logical implications of a fact and finetuning on them, and Lampinen et al. [31], who show that implication-based finetuning can even outperform in-context learning. We build on these approaches by *training* models through RL to generate more optimal finetuning data. Park et al. [32] show that prompting language models to generate question–answer (QA) pairs directly can outperform implication-style prompting. Because the SEAL framework is agnostic to the prompt and format of the self-edit data, it can also be trained to generate QA pairs or other output formats, as explored in §B.11.

**Test-Time Training.** Test-Time Training (TTT) temporarily adapts model weights based on the input the model receives [33, 34, 35, 36]. Akyürek et al. [36] show that combining TTT with ICL enables gradient-updates to outperform standard ICL in the few-shot setting. SEAL can be viewed as incorporating a round of TTT in its inner-loop optimization, leveraging TTT's efficiency relative to full-scale training to perform multiple updates and reward the generated data that yields the greatest performance gain. Although our method is trained using single-example TTT episodes, we demonstrate in the knowledge incorporation setting that it generalizes to the continued pretraining (CPT) regime—where placing data directly in context is no longer feasible.

**Reinforcement Learning for LLMs.** Reinforcement learning has played a central role in improving LLM behavior, originally through RLHF [37, 38]. More recently, RL with verifiable rewards has been applied to boost reasoning performance by optimizing the model directly for task success [39, 40, 41]. SEAL applies RL not to optimize final answers or trace revisions, but to optimize the generation of *self-edit* data that is then used for weight updates.

**Meta-Learning and Self-Modifying Systems.** SEAL embodies meta-learning principles [42, 43, 44] by learning an adaptation strategy—how to generate effective self-edits—via its outer optimization loop. The goal is to learn *how to learn* efficiently from task contexts. In reinforcement learning, meta-learning has been used to train agents that learn new tasks quickly [45, 46, 47, 48]. Sun et al. [49] similarly apply RL to learn task-specific weight modulations, offering an alternative to LoRA finetuning that is orthogonal to our approach. A natural extension of meta-learning is self-referential networks, where models modify their own parameters [50, 51]. In the domain of large language models, recent work has applied meta-learning to improve LLM adaptation [52, 53, 54, 55, 49]. Notably, Hu et al. [53] train a smaller model to output token-specific weights during finetuning, addressing a knowledge incorporation task similar to ours, while Chen et al. [54] propose a hypernetwork that generates LoRA adapters conditioned on the input, enabling dynamic and task-specific parameterization. However, SEAL offers greater generality by leveraging the model's existing generative capabilities to parametrize updates.

**Self-Improvement.** Several recent works fall under the umbrella of self-improvement or self-training. Methods such as RLAIF [56, 57] and self-rewarding language models [58, 59] use the model itself to provide reward signals, leveraging the observation that judging outputs is often easier than generating them [60]. Other recent works improve performance on mathematical tasks by using majority-vote or model confidence as reinforcement learning rewards, enabling performance improvement without access to ground-truth labels [61, 62, 63, 64, 65]. However, all of these methods are fundamentally limited by the model's current evaluation abilities and self-consistency. In contrast, we view self-improvement through interaction with external data as a more powerful and scalable path. SEAL learns how to best utilize this external data for self-improvement.

## 3 Methods

We propose Self-Adapting LLMs (SEAL), a framework that enables language models to improve themselves by generating their own synthetic data and optimization parameters ("self-edits") in response to new data. The model is trained to produce these self-edits directly through token generation with the data provided in the model's context. Self-edit generation is learned via reinforcement learning (RL) where the model is rewarded for generating self-edits (SE) that, when applied, improve the model's performance at the target task. SEAL can therefore be interpreted as an algorithm with two nested loops: an *outer RL loop*, which optimizes the self-edit generation, and an *inner update loop*, which uses the generated self-edit to update the model via gradient descent. Our method can be seen as an instance of meta-learning where we meta-learn how to generate effective self-edits.

### 3.1 General Framework

Let $\theta$ denote the parameters of the language model $\text{LM}_\theta$. SEAL operates on individual task instances $(C, \tau)$ where $C$ is a context containing information relevant to the task, and $\tau$ defines the downstream evaluation used to assess the model's adaptation. For example, in knowledge incorporation, $C$ is the passage intended to be integrated into the model's internal knowledge, and $\tau$ is a set of questions and associated answers about the passage. In few-shot learning, $C$ includes few-shot demonstrations of a

novel task, and $\tau$ is the query input and ground-truth output. Given $C$, the model generates a self-edit SE—the form of which varies by domain (see §3.2)—and updates its parameters via supervised finetuning: $\theta' \leftarrow \text{SFT}(\theta, \text{SE})$.

We optimize the self-edit generation process using reinforcement learning: the model takes an *action* (generating SE), receives a *reward* $r$ based on $\text{LM}_{\theta'}$'s performance on $\tau$, and updates its policy to maximize expected reward:

$$\mathcal{L}_{\text{RL}}(\theta_t) := -\mathbb{E}_{(C,\tau)\sim\mathcal{D}}\left[\mathbb{E}_{\text{SE}\sim\text{LM}_{\theta_t}(\cdot|C)}\left[r(\text{SE}, \tau, \theta_t)\right]\right]. \tag{1}$$

Unlike in standard RL setups, the reward assigned to a given action in our setting depends on the model *parameters* $\theta$ at the time the action is taken (since $\theta$ is updated to $\theta'$, which is then evaluated). As a result, the underlying RL state must include the policy's parameters and is given by $(C, \theta)$, even though the policy's observation is limited to $C$ (placing $\theta$ directly in context is infeasible). The implication of this is that (state, action, reward) triples collected with a previous version of the model, $\theta_{\text{old}}$, may become stale and misaligned for the current model $\theta_{\text{current}}$. For this reason, we adopt an on-policy approach, in which self-edits are sampled from—and, crucially, rewards are computed using—the current model.

---

**Algorithm 1** Self-Adapting LLMs (SEAL): Self-Edit Reinforcement Learning Loop

---

1: **Input:** $\text{LM}_\theta$, dataset $\mathcal{D} = \{(C, \tau)\}$
2: **for** outer iteration $t = 1, 2, \ldots$ **do**
3:     Sample $(C, \tau) \sim \mathcal{D}$
4:     Generate self-edit $\text{SE} \sim \text{LM}_\theta(\cdot \mid C)$
5:     Inner Loop Update: $\theta'_t \leftarrow \text{SFT}(\theta_t, \text{SE})$
6:     Evaluate: $\text{Ans} \sim \text{LM}_{\theta'_t}(\cdot \mid \tau)$
7:     Compute reward: $r \leftarrow r(\text{Ans}, \tau)$
8:     Update: $\theta_{t+1} \leftarrow \text{RL\_Update}(\theta_t, r, \text{SE})$
9: **end for**

---

We experimented with various on-policy methods such as Group Relative Policy Optimization (GRPO) [66] and Proximal Policy Optimization (PPO) [67], but found the training to be unstable. Instead, we adopt ReST$^{EM}$ [40], a simpler approach based on filtered behavior cloning—also known as "rejection sampling + SFT" [68, 69, 38, 39, 70].

ReST$^{EM}$ can be viewed as an expectation-maximization (EM) procedure: the **E-step** samples candidate outputs from the current model policy, and the **M-step** reinforces only those samples that receive positive reward through supervised finetuning. This approach optimizes an approximation of our objective (1) under the binary reward:

$$r(\text{SE}, \tau, \theta_t) = \begin{cases} 1 & \text{If on } \tau, \text{ adaptation using SE improves } \text{LM}_{\theta_t}\text{'s performance}^2 \\ 0 & \text{Otherwise} \end{cases} \tag{2}$$

More precisely, in optimizing (1), we must compute the gradient $\nabla_{\theta_t}\mathcal{L}_{\text{RL}}$. However, as we noted, the reward term $r(\text{SE}, \tau, \theta_t)$ depends on $\theta_t$ in our setup but is not differentiable. We address this by treating the reward as fixed with respect to $\theta_t$. With this approximation, the Monte-Carlo estimator for a minibatch of $N$ contexts and $M$ sampled self-edits per context becomes

$$\nabla_{\theta_t}\mathcal{L}_{\text{RL}} \approx -\frac{1}{NM}\sum_{i=1}^{N}\sum_{j=1}^{M} r_{ij}\,\nabla_{\theta_t}\log p_{\theta_t}(\text{SE}_{ij} \mid C_i) \tag{3}$$

$$= -\frac{1}{NM}\sum_{i=1}^{N}\sum_{j=1}^{M} r_{ij}\sum_{s=1}^{T}\nabla_{\theta_t}\log p_{\theta_t}(y_s^{(i,j)} \mid y_{<s}^{(i,j)}, C_i), \tag{4}$$

where $p_{\theta_t}$ denotes the model's autoregressive distribution and $y_s^{(i,j)}$ is the $s^{\text{th}}$ token of self-edit $\text{SE}_{ij}$, the $j^{\text{th}}$ sample for context $C_i$. Since sequences with $r = 0$ can be ignored in (4), we have shown that ReST$^{EM}$, with simple "SFT on good self-edits," indeed optimizes (1) under the binary reward (2) (with a stop-gradient applied to the reward term). The SEAL training loop is summarized in Alg. 1.

Finally, we note that while the implementation in this work uses a single model for both generating self-edits and learning from these self-edits, it is also possible to decouple these roles. In such a "teacher-student" formulation [71], a student model would be updated using edits proposed by a separate teacher model. The teacher would then be trained via RL to generate edits that maximize student improvement.

---

$^2$The reward may also be assigned to the single self-edit that yields the greatest improvement among sampled candidates, which we do in knowledge incorporation, rather than to all edits that yield a positive improvement.

## 3.2  Domain Instantiations

We instantiate the SEAL framework in two distinct domains: knowledge incorporation and few-shot learning. These domains were chosen to highlight two complementary forms of model adaptation: (1) the ability to integrate new information into a model's weights so that it can be recalled without relying on context (evaluated using a no-context variant of SQuAD) and (2) the ability to generalize to novel tasks after seeing only a small number of examples (evaluated using ARC).

**Knowledge Incorporation.**  Our goal is to efficiently incorporate the information provided in a passage into the model's weights. A promising recent approach involves using a language model to generate content derived from the passage, followed by finetuning on both the original passage and the generated content [29, 30, 25, 31, 32]. While the form of generated content may vary, we adopt what we consider the canonical format: *implications derived from the passage*. This approach, introduced in deductive closure training [30], converts a given context $C$ into a set of implications $\text{SE} = \{s_1, s_2, \ldots, s_n\}$ by prompting the model to "List several implications derived from the content." The output may include inferences, logical consequences, or restatements of the original passage. In §B.11, we also explore alternative prompts such as "rewrite the passage in different ways" or "rewrite in a question-answer format" and show that our method improves performance by similar or greater margins regardless of the base prompt.

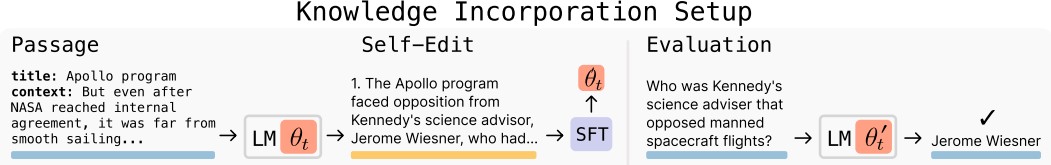

Figure 2: **Knowledge Incorporation Setup.** Given a new passage, the model generates synthetic data (the *self-edit*) in the form of "implications" of the passage. We then finetune on these outputs using LoRA. The updated model is evaluated on questions about the passage *without* access to the original text, and the resulting accuracy serves as the reward signal for reinforcement learning.

These self-generated statements form the training data for a supervised finetuning (SFT) update: we compute the standard causal language-modeling loss over each sequence $s_i$ and update the model parameters, yielding $\theta'$. Since the amount of data per update is small and the number of updates we do in total is large, we use low-rank adapters (LoRA [72]) for efficient, lightweight tuning. Finally, the adapted model $\text{LM}_{\theta'}$ is evaluated on the task $\tau$. This process is shown in Figure 2.

During RL training, the adapted model's accuracy on $\tau$ defines the reward $r$ that drives the outer RL optimization. This trains the model to restructure the passage in a way that is most effective for assimilation via finetuning.

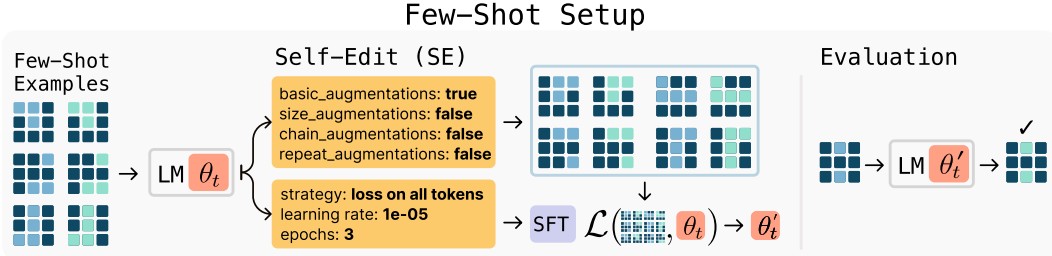

Figure 3: **Few-Shot Learning with SEAL.** Left: example ARC demonstrations. Center: the model generates a self-edit specifying augmentations and training hyperparameters. Right: the adapted model is evaluated on a held-out test input.

**Few-Shot Learning.** The Abstraction and Reasoning Corpus (ARC) [8] is a benchmark designed to test abstract reasoning and generalization from very limited examples. Each task includes a small set of input-output demonstrations and a held-out test input whose correct output must be predicted.

We adopt the test-time training (TTT) protocol of Akyürek et al. [36], where augmentations of the few-shot examples are used to perform gradient-based adaptation. Rather than relying on manually tuned heuristics for selecting augmentations and optimization settings, we train SEAL to learn these decisions. This setting tests whether SEAL can autonomously configure the adaptation pipeline—determining which augmentations to apply and what optimization parameters to use.

To implement this, we define a set of **tools**, each of which is a pre-defined function from Akyürek et al. [36] that transforms data or specifies training parameters. These include:

- **Data augmentations:** rotations, flips, reflections, transpositions, resizing operations (e.g., changing grid resolution), and chained or repeated transformations.
- **Optimization parameters:** learning rate, number of training epochs, and whether the loss is computed over all tokens or only output tokens.

The model is prompted with a task's few-shot demonstrations to generate a self-edit, which in this case is a specification of which tools to invoke and how to configure them, as shown in Figure 3. The self-edit is then applied to adapt the model via LoRA finetuning. The adapted model is evaluated on the held-out test input, and the result determines the reward for the self-edit generation.

## 4 Results

In this section we empirically evaluate SEAL across our two adaptation domains: few-shot learning and knowledge incorporation. Full training, hyperparameter, and evaluation details are provided in §A and §B.

### 4.1 Few-Shot Learning

We conduct our experiments using `Llama-3.2-1B-Instruct`, a small open-source model with no ARC-specific pretraining. Since most ARC tasks are challenging for models that have not been pretrained on ARC, we curate a subset of 11 tasks from the ARC training set and 8 from the evaluation set, filtered to ensure that they are solvable under optimal TTT configurations for a base `Llama-3.2-1B-Instruct`. While this is a small number of examples, note that Akyürek et al. [36] used the same TTT configuration for all tasks, and thus we do not need a large training set for learning a fixed self-edit. More details are included in §A.

The model is trained using ReST$^{EM}$ by sampling 15 self-edits per training task. Each self-edit is applied individually to generate 15 updated models, which are then evaluated on the corresponding held-out test example. We reinforce only those self-edits that lead to correctly adapted models, i.e., models that produce the correct output for the test input after adaptation.

After training, we evaluate the model by generating 5 self-edits per held-out evaluation task and apply each one independently. We then report the percentage of self-edits that lead to correct outputs, yielding a success rate that reflects the quality of the learned self-edit generation policy.

We compare against the following baselines:

1. **ICL (In-Context Learning):** `Llama-3.2-1B-Instruct` is prompted with the given few-shot examples using Akyürek et al. [36]'s protocol, and directly queried on the test input.
2. **TTT + Self-Edit (w/o prior RL):** `Llama-3.2-1B-Instruct` performs test-time training (TTT) using few-shot examples and synthetic augmentations, but without any prior RL to optimize which augmentations or training configurations to use.
3. **Oracle TTT:** The model performs test-time training (TTT) using the optimal human-crafted configuration from Akyürek et al. [36]. This provides an upper bound of our method.

We record results in Table 4.1. SEAL substantially improves adaptation success rate compared to baselines: 72.5% vs. 20% (with self-edits from the base model without RL training) and 0% (no adaptation), though performance remains below Oracle TTT, suggesting room for further improvement.

| Method | Success Rate (%) |
|---|---|
| ICL | 0 |
| TTT + Self-Edit (w/o prior RL) | 20 |
| SEAL | 72.5 |
| Oracle TTT | 100 |

Table 1: Few-shot Abstract Reasoning

## 4.2 Knowledge Incorporation

We experiment with `Qwen2.5-7B` on incorporating novel factual content from SQuAD passages [13]. We use the relatively simple SQuAD dataset because its passages can be fully "understood" by the base model in-context, yet the model cannot reliably answer questions about them *without* that context. We do 2 rounds of ReST$^{EM}$ with a batch of 50 contexts (see §B for further details). We compare SEAL on knowledge incorporation against the following baseline approaches:

1. **Base Model:** The pretrained model is evaluated on downstream QA tasks without any adaptation or access to the passage.
2. **Train on Passage Only:** The model is finetuned directly on the passage using the standard language modeling loss, without any synthetic data.
3. **Train on Passage + Synthetic Data:** The model is trained on the passage along with self-generated implications. This is the same setup as SEAL but without any prior RL training.
4. **Train on Passage + GPT-4.1 Synthetic Data:** The model is trained on the passage along with model-generated implications collected from GPT-4.1 via the OpenAI API.

Table 4.2 reports mean no-context SQuAD accuracy under two regimes: single-passage updating (with LoRA), and small-scale continued pretraining (with full finetuning). We run continued pretraining (CPT) experiments with $n = 200$ documents, as well as the full SQuAD validation set of $n = 2067$ documents. In the single-passage setting, finetuning directly on the passage yields a negligible gain over the frozen base model (33.5% vs. 32.7%), confirming that using the raw data alone is insufficient. Augmenting with synthetic implications generated by GPT-4.1 boosts accuracy to 46.3%, an improvement of 12.8 percentage points over the passage-only baseline. Using synthetic data produced by the base `Qwen-2.5-7B` model yields 39.7%, a 6.2-point increase. After reinforcement learning, SEAL further improves accuracy to **47.0%**, notably outperforming using synthetic data from GPT-4.1, despite being a much smaller model.

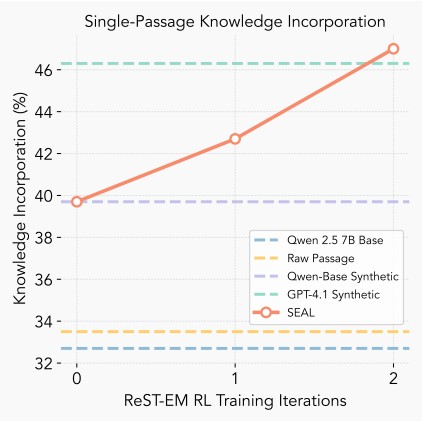

Figure 4: **Accuracy over RL iterations.** Each iteration consists of a mini-batch of 50 contexts, each with 5 sampled self-edits. SEAL surpasses GPT-4.1 synthetic data after two iterations of ReST$^{EM}$ on the no-context SQuAD set.

In the CPT setting, the model assimilates information from many passages in a single continued pretraining run. It is then evaluated on the union of all corresponding questions. In this setting, we sample 5 self-edit generations for each passage and take the aggregate synthetic dataset for continued pretraining. As shown in Table 4.2, we observe a similar ranking of methods as in the single-passage case, but with synthetic data from GPT-4.1 slightly outperforming SEAL. In the $n = 200$ setting, SEAL achieves an accuracy of 58.2%, exceeding its single-passage performance. We attribute this gain to the aggregation of multiple self-edit generations. Overall, the strong continued pretraining results of SEAL suggest that the self-editing policy generalizes *beyond* the original RL setup of creating synthetic data in a single generation for a single passage.

Figure 4 tracks accuracy after each outer RL iteration. Two iterations suffice for SEAL to overtake GPT-4.1 data; subsequent iterations yield diminishing returns, suggesting that the policy quickly converges to an edit style that distills the passage into easily learnable atomic facts (see qualitative examples in Figure 5). All results use tuned hyperparameters (see §B).

| Method | Single Passage (n = 1; LoRA) | Continued Pretraining (n = 200; full-FT) | Continued Pretraining (n = 2067; full-FT) |
|---|---|---|---|
| Base model | 32.7 | 32.7 | 29.0 |
| Train on Passage | 33.5 | 36.0 | 31.2 |
| Train on Passage + Synthetic | 39.7 | 50.6 | 43.4 |
| Train on Passage + GPT-4.1 Synthetic | 46.3 | **59.4** | **49.2** |
| **SEAL** | **47.0** | 58.2 | 46.4 |

Table 2: Knowledge Incorporation Performance Across Passage Settings.

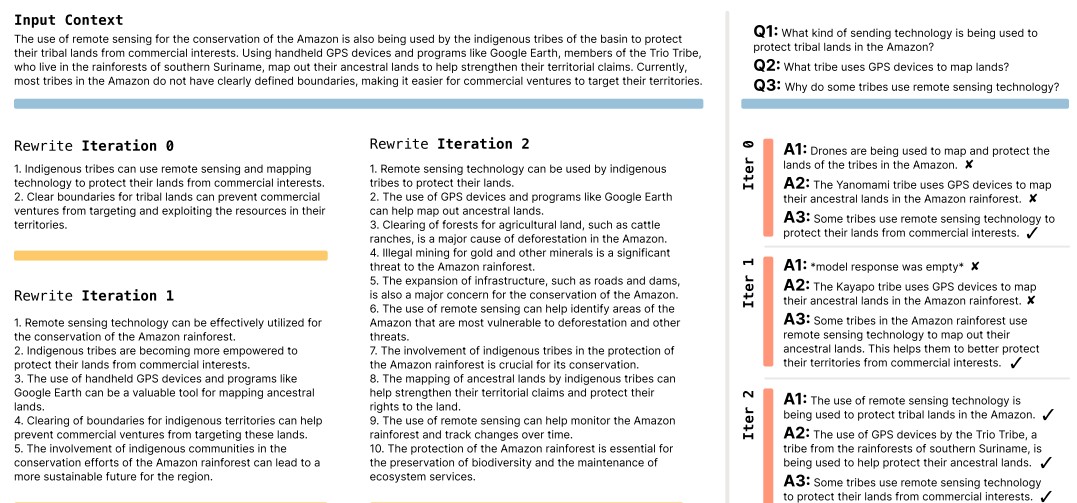

Figure 5: **Example Knowledge Incorporation Self-Edits Across RL Iterations.** In this example, we see how RL leads to the generation of more detailed self-edits, which in turn results in better performance. While the progression is clear in this case, the differences across iterations are sometimes more subtle in other examples. We show in §B.11 that prompting for longer self-edits is effective, and that RL training further improves performance by a similar margin.

# 5   Limitations

**Catastrophic forgetting.**  One key motivation we had for enabling language models to self-edit is to move towards the ultimate goal of continual learning—allowing models to incorporate new information over time, whether through agentically interacting with an environment or through standard training. While our earlier experiments assess how well SEAL adapts to individual edits in isolation, a more ambitious goal is to support *sequences* of edits: can the model adapt to new information repeatedly while preserving prior knowledge?

This question relates directly to the challenge of *catastrophic forgetting* [73, 74], where new updates interfere destructively with past learning. We do not explicitly optimize for retention in our current training setup, but we aim to establish a baseline for how well SEAL handles sequential self-edits without dedicated mechanisms for handling catastrophic forgetting. To test this, we simulate a continual learning setting in the knowledge incorporation domain. The model receives a stream of test passages, each triggering a new self-edit. After each update, we

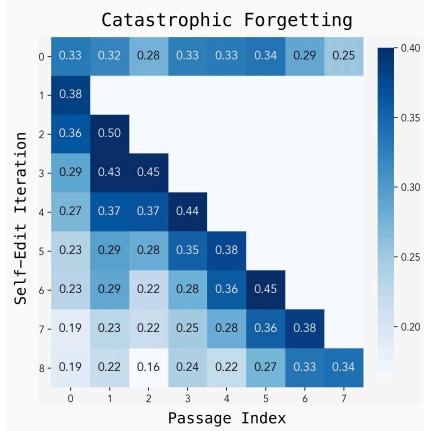

Figure 6: **Catastrophic forgetting from continual self-edits.** We sequentially update the model on new passages and track degradation on prior tasks. Entry-wise standard errors are reported in §B.6.

re-evaluate the model on all previously seen tasks to measure retention. This setup tests the model's ability to integrate new edits without forgetting earlier ones.

As shown in Figure 6, performance on earlier tasks gradually declines as the number of edits increases, suggesting that SEAL is still susceptible to catastrophic forgetting. Still, it can perform multiple updates without complete collapse, indicating possibility for improvement. Future work could enhance this ability through reward shaping [75, 76, 77] to penalize regressions on earlier tasks, or by integrating continual learning strategies such as null-space constrained edits [78] or representational superposition [79]. In addition, since RL has been shown to forget less than SFT, SEAL's inner loop could also employ RL instead of SFT [80].

**Computational overhead.**   The TTT reward loop is significantly more computationally expensive than other reinforcement learning loops used with LLMs. For instance, reward signals based on human preferences typically involve a single model forward pass, and those using verified solutions may rely on simple pattern matching (e.g., regex). In contrast, our approach requires finetuning and evaluating an entire model to compute the reward—each self-edit evaluation takes approximately 30–45 seconds, introducing substantial overhead (see §B.5).

**Context-dependent evaluation.**   Our current instantiations assume that every context is paired with an explicit downstream task: few-shot demonstrations arrive with a held-out query pair, and each passage comes bundled with reference QA. This coupling simplifies reward computation but prevents RL training of SEAL from scaling to unlabeled corpora. A potential solution is to let the model generate not only self-edits but also its own evaluation questions—e.g., draft QA items or synthetic test cases for each passage—while the original content is still in context. These model-written queries could provide the immediate supervision required for reinforcement learning, broadening applicability to general training domains where external question-and-answer sets are unavailable.

## 6    Discussion and Conclusion

Villalobos et al. [81] project that frontier LLMs will be trained on all publicly available human-generated text by 2028. We argue that this impending "data wall" will necessitate the adoption of synthetic data augmentation. Once web-scale corpora are exhausted, progress will hinge on a model's capacity to *generate its own high-utility training signal*. A natural next step is to meta-train a dedicated SEAL synthetic-data generator model that produces fresh pretraining corpora, allowing future models to scale and achieve greater data efficiency without relying on additional human text.

We can imagine a future in which LLMs can ingest new data, such as academic papers, and generate large quantities of explanations and implications for themselves using their existing knowledge and reasoning with the in-context data. This iterative loop of self-expression and self-refinement could allow models to keep improving on rare or underrepresented topics even in the absence of additional external supervision.

In addition, while modern reasoning models are often trained with RL to generate chain-of-thought (CoT) traces, SEAL could offer a complementary mechanism, allowing the model to learn when and how to update its own weights. These two approaches could synergize: the model may choose to perform weight updates mid-reasoning to guide its current trajectory, or after completing reasoning to distill key insights into its parameters—improving future inference through internalized learning.

This continual refinement loop is also promising for building agentic systems—models that operate over extended interactions and adapt dynamically to evolving goals. Agentic models must incrementally acquire and retain knowledge as they act. Our approach supports such behavior by enabling structured self-modification: after an interaction, the agent could synthesize a self-edit which triggers a weight update. This could allow the agent to develop over time, aligning its behavior with prior experience and reducing reliance on repeated supervision.

SEAL demonstrates that large language models need not remain static after pretraining: by learning to generate their own synthetic self-edit data and to apply it through lightweight weight updates, they can autonomously incorporate new knowledge and adapt to novel tasks. Looking ahead, we envision extending the SEAL framework to pretraining, continual learning, and agentic models, ultimately enabling language models to self-learn and scale in a data-constrained world.

## Acknowledgments and Disclosure of Funding

We would like to thank Shivam Duggal, Idan Shenfeld, Seungwook Han, Jeremy Bernstein, Akarsh Kumar, Linlu Qiu, Juno Kim, Brian Cheung, Moritz Reuss, Ayush Sekhari, Zhang-Wei Hong, Mehul Damani, Leshem Choshen, and Ryan Yang for their valuable discussions and feedback. We acknowledge support from ARO MURI grant number W911NF-23-1-0277. This research was also partly sponsored by the United States Air Force Research Laboratory and the United States Air Force Artificial Intelligence Accelerator and was accomplished under Cooperative Agreement Number FA8750-19- 2-1000. The views and conclusions contained in this document are those of the authors and should not be interpreted as representing the official policies, either expressed or implied, of the United States Air Force or the U.S. Government. The U.S. Government is authorized to reproduce and distribute reprints for Government purposes, notwithstanding any copyright notation herein. We acknowledge the MIT Office of Research Computing and Data for providing high performance computing resources that have contributed to the research results reported within this paper. This research was also partly supported by the Stevens Fund for MIT UROP research and by the MIT-IBM Watson AI Lab.

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

# A Experimental Details: Few-shot Learning

## A.1 Model and Setup

For the few-shot learning experiments, we use `Llama-3.2-1B-Instruct` [3] as the base language model. Since this model has no specialized training on ARC, its ability to solve ARC tasks is limited. To enable controlled evaluation, we curated a small set of ARC problems from the training and evaluation splits that are solvable with optimal TTT hyperparameters.

**Training Set:** We selected 11 ARC tasks from the training set as the environment for RL optimization. **Evaluation Set:** We selected 8 distinct ARC problems from the evaluation set for measuring generalization performance. These 8 were explicitly filtered for being amenable to TTT out of the full evaluation set.

These sets were chosen to isolate the effect of self-edit learning rather than general ARC ability.

## A.2 Training Procedure

We trained SEAL using a single outer loop of reinforcement learning on the 11 training problems. For each problem, the model generated 15 self-edit configurations consisting of:

- **Data augmentation strategy:** Controls whether to include variations such as basic examples, size-based augmentations, chained reasoning, and repeated inputs: `include_basic`, `include_size`, `include_chain`, `include_repeat`.

- **Optimization configuration:** Specifies hyperparameters such as learning rate, number of epochs, and loss function type (e.g., whether to compute loss over all tokens or only the final output tokens).

Each configuration was evaluated via test-time training (TTT), and assigned a binary reward: 1 if the adapted model produced the correct solution, 0 otherwise using Akyürek et al. [36]'s evaluation protocol. To encourage efficient adaptation, we discarded configurations requiring more than 375 training steps, retained only configurations with reward 1 and acceptable cost for LoRA-based SFT.

SFT for TTT was done with the following hyperparameters:

| Parameter | Value |
|---|---|
| LoRA rank | 128 |
| LoRA alpha | 16 |
| Learning rate | N/A (model chooses it) |
| `num_train_epochs` | N/A (model chooses it) |

LoRA adapters were inserted into the following modules: `q_proj`, `v_proj`, `gate_proj`, `down_proj`, `up_proj`

For the RL SFT we used the following hyperparameters:

| Parameter | Value |
|---|---|
| `lora_rank` | 16 |
| `lora_alpha` | 16 |
| `num_train_epochs` | 8 |
| `per_device_train_batch_size` | 5 |
| `learning_rate` | 5e-5 |

When generating the self-edits, we add the following prompt as a suffix to the few-shot examples.

You are configuring a model training pipeline by selecting from predefined tools.
You must make two decisions:
1. **Data Generation Tools** — For each of the following, choose true or false:
   - use_basic_augmentations
   - use_size_augmentations
   - use_chain_augmentations
   - use_repeat_augmentations

2. **Training Configuration** — Choose one of:
   - `"train_using_all_tokens"`
   - `"train_using_output_tokens"`

Also specify:
   - learning_rate (float)
   - num_train_epochs (integer)

**Output Format**

Respond with a valid JSON object. Do not include any explanation, markdown, or extra text.
Use lowercase `true/false` for booleans and ensure correct JSON syntax.
Example output:

```
{
  "data_generation": {
    "use_basic_augmentations": ...,
    "use_size_augmentations": ...,
    "use_chain_augmentations": ...,
    "use_repeat_augmentations": ...
  },
  "training": {
    "strategy": ...,
    "learning_rate": ...,
    "num_train_epochs": ...
  }
}
```

## A.3 Evaluation Details

For each of the 8 held-out evaluation tasks, the model generated 5 self-edit configurations, yielding a total of 40 configurations. Success was measured as the percentage of configurations that led to correct outputs after adaptation. We followed the evaluation protocol from Akyürek et al. [36].

For the Oracle TTT we used the following configs:

| Parameter | Value |
|---|---|
| lora_rank | 128 |
| lora_alpha | 16 |
| num_train_epochs | 2 |
| batch_size | 2 |
| learning_rate | 1e-4 |

## A.4 Compute Resources

We performed all training runs on a single A100, H100, or H200. Each TTT per problem requires between half a minute to a few minutes, which is also why we limited the number of samples for ReST$^{EM}$ and additionally limited the number of gradient steps allowed per self-edit TTT. Overall ReST$^{EM}$ took around 2-3 hours.

# B  Experimental Details: Knowledge Incorporation

## B.1  Model and Setup

We use the `Qwen-2.5-7B` base model [5] in the knowledge incorporation experiments. We repurpose the SQuAD dataset v1.1 [13] for the task of answering questions *without* the passage in-context. We use the training set for RL training and a 200-article subset of the evaluation set for evaluation. Within the training set and evaluation set, there are some overlapping topics of passages, but there is no overlap between these sets, so we can be sure that there is no data contamination of the test passages due to RL training.

## B.2  RL Training Procedure

We run 2 rounds of ReST$^{EM}$ training [40]. On each round, we take a batch of 50 context-questions-answers triples from the SQuAD training set. For each context, we sample 5 self-edit generations at temperature 1. We evaluate each self-edit over 3 random seeds, training on the sequences and then evaluating the updated model on the corresponding questions. We average each generation's results over 3 seeds and then keep the single best generation for each of the 50 contexts. Finally, to finish the round of ReST$^{EM}$, we perform supervised finetuning on the 50 resulting prompt-completion pairs.

Supervised finetuning here is done with batch size of 10, for 2 epochs, with learning rate 3e-4, using LoRA [72] with rank 64 and alpha 128, applied to all MLP and attention projection layers.

## B.3  Synthetic Data Generation and Finetuning Details

In all models, we generate synthetic data by prompting to generate implications of the passage:

> Let's read the following passage and produce a list of implications derived directly or indirectly from the content.
>
> Passage:
> {passage}
>
> Implications:

We then take the resulting generated sequence. In the single-passage case, we split it by newlines into a set of training documents. In the multi-passage case, we use the full generated sequence as a single training document. In the case of synthetic data from GPT-4.1 (`gpt-4.1-2025-04-14`), an instruct-model, we additionally have the following rule: If the second line begins with a "1." then we omit the first line from the training set. This is because we found that the first line often contained filler text (e.g. "Sure, here is the list of implications:").

We then use the following training hyperparameters:

Table 3: Single-Passage Knowledge Incorporation Hyperparameters

| Parameter | Search Space |
|---|---|
| LoRA Rank ($r$) | [**32**, 64] |
| LoRA Alpha ($\alpha$) | [32, **64**] |
| Learning Rate | [1e-4, 3e-4, 5e-4, **1e-3**, 2e-3] |
| Epochs | [1, 5, **10**, 15, 20] |
| Batch Size | [**1**, 4] |

In the multi-passage $n = 200$ case, we sample 5 self-edit completions for each passage and take the aggregate dataset of all self-edits across all passages to train on.

To answer the corresponding questions, we use the following prompt:

Table 4: Multi-Passage Knowledge Incorporation Hyperparameters

| Parameter | Search Space |
|-----------|-------------|
| LoRA Rank ($R$) | [**32**, 64] |
| LoRA Alpha ($\alpha$) | [32, **64**] |
| Learning Rate | [1e-4, 3e-4, 5e-4, **1e-3**, 2e-3] |
| Epochs | [1, **3**, 5] |
| Batch Size | [1, 4, **8**, 16] |

---

Let's answer a question directly and concisely.
Question: {question}
Answer:

---

## B.4 Evaluation Details

We evaluate on a 200-passage subset of the SQuAD evaluation set, consisting of a combined 974 evaluation questions (roughly 5 corresponding to each passage). The pipeline of generating synthetic data and finetuning on it is the same as above. For automated grading, we use `gpt-4.1-2025-04-14` [82] via the OpenAI API with greedy decoding.

The grading prompt is as follows:

---

You are a grading assistant. Your job is to determine whether a student's answer correctly answers the question based solely on the provided gold answer. Do not use any outside knowledge. The student answer can include additional information, but it must at least fully convey the gold answer and must not contradict it. Ignore style, phrasing, or extra details that do not affect correctness. Respond ONLY with 'yes' or 'no'.

Question: {question}
Gold answer: {gold}
Student answer: {pred}
Is the student answer correct based solely on the gold answer? Respond 'yes' or 'no'.

---

## B.5 Compute Resources

All experiments are performed on 2×H100 or 2×H200. We use DeepSpeed ZeRO-3 [83] for SFT in ReST$^{EM}$ training. We use vLLM [84] for efficient inference. The most compute-intensive portion of our training and evaluation is the E-step of ReST$^{EM}$ training, where the model generates completions and is graded through the inner-loop process of finetuning and running inference. Doing a single round requires a batch of 50 passages over 5 completions and 3 runs per completion, meaning 750 inner loop iterations. This takes about 6 hours on 2×H100s.

## B.6 Standard Error of the Mean in Catastrophic Forgetting Experiment

The standard errors of the mean (SEM) for each entry in Figure 6 is shown below in Table B.6.

## B.7 Scaling Model Size

We further experimented with the 3B-parameter Qwen variant, with the same single-passage setup as in Figure 4. The results are given in Table 6.

To compare the benefit of SEAL over using self-edits generated by the base model, we compute the ratio of SEAL's improvement over the base model to the improvement from base model self-edits. This ratio is $1.75\times$ for the 3B model and $2.04\times$ for the 7B model. The relative improvement is greater for the 7B model, which provides some evidence that not only are stronger base models more effective at leveraging synthetic data for self-adaptation, but reinforcement learning may have compounding

Table 5: Entrywise standard errors of the mean (SEM) across continual self-edits experiment.

|   | 1 | 2 | 3 | 4 | 5 | 6 | 7 | 8 |
|---|---|---|---|---|---|---|---|---|
| 0 | 0.0306 | 0.0315 | 0.0263 | 0.0318 | 0.0297 | 0.0370 | 0.0310 | 0.0284 |
| 1 | 0.0273 | 0.0000 | 0.0000 | 0.0000 | 0.0000 | 0.0000 | 0.0000 | 0.0000 |
| 2 | 0.0305 | 0.0277 | 0.0000 | 0.0000 | 0.0000 | 0.0000 | 0.0000 | 0.0000 |
| 3 | 0.0277 | 0.0358 | 0.0406 | 0.0000 | 0.0000 | 0.0000 | 0.0000 | 0.0000 |
| 4 | 0.0272 | 0.0303 | 0.0337 | 0.0320 | 0.0000 | 0.0000 | 0.0000 | 0.0000 |
| 5 | 0.0296 | 0.0342 | 0.0290 | 0.0298 | 0.0319 | 0.0000 | 0.0000 | 0.0000 |
| 6 | 0.0289 | 0.0334 | 0.0271 | 0.0258 | 0.0320 | 0.0337 | 0.0000 | 0.0000 |
| 7 | 0.0255 | 0.0313 | 0.0264 | 0.0253 | 0.0309 | 0.0331 | 0.0363 | 0.0000 |
| 8 | 0.0237 | 0.0307 | 0.0211 | 0.0267 | 0.0273 | 0.0271 | 0.0358 | 0.0263 |

Table 6: Model Size Scaling Performance (%).

| Model | Base Model (No Training) | Base Model Self-Edit | SEAL |
|---|---|---|---|
| Qwen2.5-3B | 25.1 | 31.9 | **37.0** |
| Qwen2.5-7B | 32.7 | 39.7 | **47.0** |

benefits as model capacity increases. We acknowledge that it is hard to draw conclusions though without actually scaling up further.

## B.8 Comparison to Generative Adapter

We additionally compared with Generative Adapter [54], a hypernetwork approach that generates LoRA weights from context, using our evaluation setup. Table 7 reports results for both single-passage ($n=1$) and continued pretraining ($n = 200$). We use the Mistral-7B-based model [85] for Generative Adapter, since that was the closest model for comparison. All values are on the same evaluation set, but CPT batches updates over all documents while single-passage trains and evaluates an adapter separately for each document. Generative Adapter achieves strong performance in the $n=1$ case, but underperforms SEAL in the CPT setting. SEAL's parameterization of weight updates through synthetic data generation allows reuse of generated data for CPT, application to arbitrary base models, and flexibility to learn updates from diverse interaction types beyond LoRA finetuning.

Table 7: SEAL vs. Generative Adapter Performance (%).

| Model | Base | Single-passage ($n=1$) | CPT ($n = 200$) |
|---|---|---|---|
| SEAL | 32.0 | 47.0 | **58.2** |
| Generative Adapter | 24.4 | **66.8** | 28.0 |

We note that parameterizing weight updates via synthetic data generation rather than directly predicting LoRA weights has several advantages: (1) generated data can be reused for CPT or applied to arbitrary base models, (2) models can leverage reasoning and restructuring as document scale and complexity grow, and (3) the framework is not restricted to LoRA finetuning, allowing for many different update types, including those arising from environment or user interactions. By contrast, it is unclear how hypernetwork-based approaches would scale to such settings, while next-token prediction on generated data naturally exploits a model's in-context learning capabilities.

## B.9 Comparison to Entigraph

We additionally compare SEAL to Entigraph [25] in the Synthetic Continued Pretraining (SCPT) setting on SQuAD. Results for both 200 and 2067 passages are shown in Table 8. SEAL uses the same 5 synthetic data generations per document. For Entigraph, we sample 5 synthetic data generations involving pairs and 5 triplets of entities per document. Entigraph with all 10 synthetic data generations sampling is competitive with SEAL, especially at the larger scale. These results

suggest that RL-trained self-edits and structured heuristic methods are both strong approaches for synthetic data generation.

Table 8: Synthetic Continued Pretraining (SCPT) on SQuAD (no passage in context). Best in each column is bolded.

| Method | Continued Pretraining (n=200) | Continued Pretraining (n=2067) |
|---|---|---|
| SEAL | **58.2** | 46.4 |
| Entigraph (pairs) | 46.2 | 38.6 |
| Entigraph (pairs+triples) | 56.0 | **48.6** |

## B.10    Proxy Reward

We experiment with replacing the inner loop with a proxy reward based on a human-crafted rubric with 4 categories: length, diversity, quality, and correctness. A GPT-4.1 grader scores each category on a 1-5 scale, and the sum of these scores is used as the RL reward. Table 9 reports final results and RL training times.

Table 9: Full Reward vs. Proxy Reward Performance (%).

| Model | Base | Post-RL | Time |
|---|---|---|---|
| SEAL | 32.0 | **47.0** | $\approx$6 hr |
| SEAL w/ Proxy-Reward | 32.0 | 45.6 | $\approx$**5 min** |

While further tuning of the rubric or metric design could strengthen the reward signal, the advantage of the full SEAL loop is that no such manual specification is required—the model directly learns which edits improve its own performance. Both approaches appear promising for scaling to larger model sizes and compute budgets: proxy metrics offer dramatically lower cost, and with refinement, they may even surpass the "true" reward of directly optimizing for post-finetuning performance.

## B.11    Prompting

Recent works have shown that reinforcement learning baselines and outcomes can be highly sensitive to prompting. We experiment with 6 additional self-edit prompts in the knowledge-incorporation setting. The seven prompts—`implications`, `implications-long`, `implications-very-long`, `implications-chain-of-thought`, `rewrite`, `self-qa`, and `no-prompt`—are shown below. All results in the main content of the paper used the `implications` prompt, which we consider to be the most prototypical [30, 31]. However, prior work has found prompts involving rewriting or generating question-answer pairs can be more effective, as discussed in §2.

Furthermore, as we see qualitatively in Figure 5, RL appears to have dramatically increased the length of the response of the example. We therefore experiment with prompting for longer generations with `implications-long` and `implications-very-long` to test if we can achieve similar gains through prompting alone.

The results are shown in Table 10. Notably, the baselines for `implications-long` and `rewrite` the RL-trained version of `implications`. However, using these prompts as the base of RL training yields even greater improvements. In all cases, ReST$^{EM}$ enhanced performance by roughly 6 to 11 percentage points.

Here, "Chain-of-thought-eval" refers to having the model reason before *answering* the questions (letting the model "pull out" information from its weights), rather than chain-of-thought before generating synthetic data, which is done with the base `implications` prompt. However, we did not notice a substantial difference in our setting when chain-of-thought was applied, whether before answering and before writing synthetic data.

Letting the model "determine its own" self-edit format, with `no-prompt`, was not able to achieve the same results as predefined prompting formats in our experiments, achieving only 18.9% after 2 rounds of training.

| Method | Original | Round 1 | Round 2 | gpt-4.1 synthetic |
|---|---|---|---|---|
| No self-edit | 33.5 | – | – | – |
| Implications | 39.7 | 43.7 | **47.0** | 46.3 |
| Implications-long | 49.3 | 52.4 | **54.4** | 54.1 |
| Implications-very-long | 45.0 | 51.5 | **52.1** | 40.9 |
| Rewrite | 49.4 | 55.3 | **55.6** | 54.4 |
| Self-QA | 37.3 | 42.8 | **48.7** | 39.2 |
| No-Prompt | 13.8 | 12.7 | 18.9 | **28.6** |
| Implications-chain-of-thought | 38.7 | – | – | – |
| *Chain-of-thought-eval* | 37.8 | – | – | – |

Table 10: Performance across 2 rounds of ReST$^{EM}$ RL training on various prompts in the single-document knowledge incorporation setting. The *gpt-4.1* column reports performance using synthetic data generated by gpt-4.1 with the corresponding prompt format.

The five prompts are shown below.

```
implications
```

Let's read the following passage and produce a list of implications derived directly or indirectly from the content.

Passage:
{passage}

Implications:

```
implications-long
```

Let's read the following passage and produce a long list of implications derived directly or indirectly from the content.

Passage:
{passage}

Implications:

```
implications-very-long
```

Let's read the following passage and produce a very long list of implications derived directly or indirectly from the content.

Passage:
{passage}

Implications:

```
implications-chain-of-thought
```

Let's read the following passage, think step by step, and then produce a list of implications derived directly or indirectly from the content. We should first generate a "Thought Process" and then "Implications"

Passage:
{passage}

Thought Process:

```
no-prompt
```

{passage}

```
rewrite
```

Let's read the following passage and rewrite it in a few different ways, each one separated by a newline.

Passage:
{passage}

Rewritten passages:

```
self-qa
```

Let's read the following passage and rewrite it in a question-answer format.

Passage:
{passage}

Question 1:

**Note:** For `self-qa`, we apply additional formatting so that training documents consist of question–answer pairs, rather than using our standard approach of splitting by newline characters. Specifically, we split the output using occurrences of "Question n:" instead of newlines.

