# OpenReview forum: "Self-Adapting Language Models"
_NeurIPS.cc/2025/Conference — NeurIPS 2025 poster_

### Official Review · Reviewer_oBBv · 2025-06-20

**Clarity:** 3
**Significance:** 3
**Originality:** 3
**Rating:** 5
**Confidence:** 4

**Summary:**

This paper proposes SAEL, a novel algorithm for training LLMs to adapt themselves to particular tasks with RL. SEAL works by sampling outputs produced by the LLM which parameterize novel (synthetic) data that the LLM is subsequently finetuned on with LoRA. The LLM is trained with RL to output synthetic training data that improves its performance on the task at hand.

SEAL is evaluated on a tool-use setting for ARC, as well as in a QA setting with natural language. In both cases SEAL works well. The authors also showcase example of the edits, and highlight some important limitations.

**Questions:**

* How do the gains from SAEL scale with the size/quality of the base LLM?
* How does this technique compare to RL/reasoning methods? Simply comparing SEAL to a CoT approach without training would add value here.
* Are there any qualitative differences between the self edits that improve Knowledge Incorporation and those that do not? From Figure 5 it seems that the LLM learns to output longer, more specific edits, but this seems pretty subtle. Would love to see some more detailed examples of successful vs unsuccessful edits.


Minor remarks:
* It's not obvious from the text that the base LLM used in the QA experiment is Qwen-2.5-7B. It's stated in the appendix, but should be clear in the main text.

Overall I think the idea and algorithm is nice and makes a lot of sense. Unfortunately the evaluation of the algorithm is too superficial for me to recommend more than borderline accept. However, if the authors can address the questions above I am willing to increase my score.

**Ethical Concerns:**

["NO or VERY MINOR ethics concerns only"]

**Final Justification:**

The method introduced paper is intuitive and original. I had some doubts about the evaluation, particularly how SEAL compares to more conventional CoT prompting, which the authors have addressed in the rebuttal.

**Limitations:**

Yes

**Paper Formatting Concerns:**

No concerns.

**Quality:**

2

**Strengths And Weaknesses:**

Strengths:
* The algorithm is novel and interesting.
* SEAL also looks to work quite well in the domain tested.
* Algorithm is fairly simple to understand.



Weaknesses:
* In SEAL, the algorithm generates data for itself, which means that if it's not very knowledgeable about a topic in the first place, it will be challenging to generate edits that improve performance. In this case, training on synthetic data from a frontier model like GPT4.1 seems to have advantages.
* I think a slightly more extensive evaluation of the algorithm is necessary for publication. Particularly, I wonder how the performance scales with the quality of the base LLM used. Are stronger models also better at generating synthetic training data for themselves? I think rerunning the QA experiment with differently sized Qwen models could answer this question.
* How does SEAL compare to Chain of Thought (CoT) reasoning? It seems plausible that CoT could pull out relevant pieces of information that the LLM knows and which could improve performance too when added to the context.

---

> ### Author Rebuttal · Authors · 2025-07-31
>
> We thank the reviewer for their constructive feedback! We will address the proposed weaknesses and questions.
>
> **Response to Weaknesses:**
>
> 1. **Smaller Models Might Not Have Enough Knowledge.** We agree that SEAL’s ability to generate useful self-edits is tied to the underlying capabilities of the base model. However, our results show that even small 7B models can substantially benefit, outperforming using data generated by GPT-4.1, despite starting with much lower baseline knowledge. We agree though that starting with a stronger base model would likely greatly improve the ability to self-edit, as scaling increases generation length and the ability to synthesize "new" information in context, given the original piece of data.
>
> 2. **Scaling Model Size.** To further explore this, we conducted an additional scaling experiment across different Qwen model sizes (3B and 7B). We observed that the difference in performance between the base model and the SEAL model increased with the larger model, indicating that stronger base models are indeed better at generating synthetic data for self-adaptation, and that the benefits of RL increase as well. We will include these results in the revised paper and have provided them at the bottom of this response.
>
> 3. **Chain-of-Thought.** We’ve run experiments incorporating CoT in two different ways. First, in producing a self-edit, we experiment with having the model write a CoT before the generation of synthetic data (see table under response to reviewer nQa9). We found little difference in performance in our document QA domain (39.7% vs 38.7% with base model), likely due to the fact that generating implications of our relatively simple documents require little in-context “reasoning.” In addition, we ran an experiment where given the questions during evaluation, we only do CoT with no training on the original document, to see if CoT can “pull out relevant pieces of information from the LLM,” as you suggested. We believe this intuition makes sense, but did not find any substantial difference in our setting (37.8%).
>
> **Response to Questions:**
>
> 1. **Scaling Model Size.** See response to weakness 2.
>
> 2. **Chain-of-Thought.** See response to weakness 3.
>
> 3. **Qualitative Difference in Self-Edits.** We appreciate the suggestion to include more qualitative analysis of self-edits. Our initial inspection of edits did not reveal clear differences beyond length, specificity, and perhaps diversity (as noted in Figure 5), but we agree that more illustrative examples would be helpful. As we note in our response to reviewer nQa9, length clearly does not fully explain the difference. We will expand the appendix with additional qualitative examples and will provide a link to the repo containing all of the data. Furthermore, we show in our response to reviewer hhPg that a proxy reward based on these attributes can achieve good results with much less computation.
>
> **Response to Minor Remarks:**
>
> 1. **Clarify Paper Text.** We will revise the paper to clearly state that the base LLM used in the QA experiments is Qwen2.5-7B.
>
> ## Additional Experiment: Scaling Model Size
>
> We rerun our experiments with the 3B-parameter Qwen variant. The results are given below.
>
> |      Model | Base Model (No training) | Base Model Self-Edit |   SEAL |
> | ---------: | -----------------: | ---------------: | --------: |
> | Qwen2.5-3B |              25.1% |            31.9% | **37.0%** |
> | Qwen2.5-7B |              32.7% |            39.7% | **47.0%** |
>
> We can calculate the relative improvement of SEAL over the base model compared to using self-edits produced by the base model over the base model:
>
> $\\begin{align*}&\\text{3B: }\\frac{37.0−25.1}{31.9-25.1}=1.75\\\\ &\\text{7B: }\\frac{47.0−32.7}{39.7−32.7}=2.04.\\end{align*}$
>
> The relative improvement is greater for the 7B model, which provides some evidence that not only are stronger base models more effective at leveraging synthetic data for self-adaptation, but reinforcement learning may have compounding benefits as model capacity increases. We acknowledge that it is hard to draw conclusions though without actually scaling up.

---

> > ### Comment · Reviewer_oBBv · 2025-08-03
> >
> > Thanks for implementing the experiments I suggested. I'm increasing my score to 5.

---

### Official Review · Reviewer_hhPg · 2025-06-28

**Clarity:** 3
**Significance:** 3
**Originality:** 2
**Rating:** 4
**Confidence:** 3

**Summary:**

The paper proposes SEAL, a framework using RL to guide models in generating useful self-edit data.

**Questions:**

1. Beyond Figure 5, is there systematic analysis of what makes a `good` self-edit (e.g., atomicity, abstraction)?
2. Can the inner-loop reward signal be approximated more efficiently (e.g., proxy models) for scalability, especially in pretraining?

**Ethical Concerns:**

["NO or VERY MINOR ethics concerns only"]

**Final Justification:**

The rebuttal resolved my main concerns.

**Limitations:**

yes

**Quality:**

3

**Strengths And Weaknesses:**

Strengths:
1. The idea is interesting and intuitive, addressing the limitation of static and passive learning in LLMs.
2. SEAL succeeds in two distinct tasks (i.e., knowledge integration and few-shot abstract reasoning) demonstrating its versatility. In knowledge integration, SEAL's synthetic data outperforms GPT-4.1, highlighting the potential of tailored learning materials. On ARC tasks, success rates improved from 0% to 72.5%.
3. The paper is well-structured, with clear explanations and helpful figures (Figure 1, 2, 3).

Weaknesses:
1. The generalization of learned policy is unclear. The framework optimizes predefined self-edit formats (e.g., implications) but may not adapt to entirely different formats (e.g., code, math). Its robustness on harder, uncontrolled tasks (e.g., full ARC benchmark) is uncertain.
2. Experiments are limited to small-scale scenarios (single or 200 paragraphs), far from real-world continuous learning or pretraining. Larger-scale tests are needed to assess interference and shortcut learning risks.
3. Binary rewards (success/failure) may encourage hacking—generating edits tailored to specific evaluation sets rather than genuine understanding. Many real-world scenarios lack predefined evaluation metrics.
4. Fairness in comparisons: GPT-4.1 used a generic prompt; optimized prompts might improve its performance.
5. The computational cost of SEAL is high (6 hours per RL round).
5. Only smaller models were tested. Would larger models (e.g., 70B) benefit less from SEAL due to better inherent self-editing ability?
6. Flexibility is overstated. Self-edit formats are predefined (e.g., implications, JSON), making SEAL more like automated prompt engineering than true `learning to learn`.

---

> ### Author Rebuttal · Authors · 2025-07-31
>
> We thank the reviewer for their thought-out response! We address the proposed weaknesses and questions below.
>
> **Response to Weaknesses:**
>
> 1. **Generalization.** The QA task is meant to simulate general training data. We focused on document QA for its ease of verification, but concede that RL training on only this likely would not generalize to many other types of LM training data. While we don't have experiments specifically on math/code, we believe the method would still work.
>
> 2. **Larger-Scale Experiments.** We have to start somewhere! We've added a larger scale CPT result (all 2067 passages of the validation set, with full finetuning rather than LoRA) to the paper, discussed in the response to reviewer nQa9. This shows the self-edit policy scales well as we increase the amount of ingested data, so long as they are applied in a batch. As we point out in our limitations section though, sequential self-edits (without maintaining optimizer states, replay, etc.) remain prone to catastrophic forgetting.
>
> 3. **Reward Hacking.** We believe rewarding the ability to answer questions about a passage, as well as applying a few-shot rule to a novel input, are fairly robust to reward hacking. Note that these questions are not seen by the model while generating the self-edit. Is there a specific failure case you imagine?
>
> 4. **Prompting.** We've added a much more in-depth analysis across different prompts (see response to review nQa9).
>
> 5. **Computational Cost.** Computational cost is indeed high, but this is a fixed cost. After training the model to perform self-edits (or training a teacher model to generate strong synthetic data), we can then use this model for data generation indefinitely without further expensive meta-learning.
>
> 6. **Scaling Model Size.** We didn't have the capacity to run experiments on 70B models (especially with our meta-learning RL setup). However, we hypothesize that scaling will *greatly improve* self-editing ability (both before and after RL). In particular, the ability to generate synthetic data given a document is governed by the model's in-context learning ability, and its ability to generate long sequences of data conditioned on the original passage. Scaling increases context/generation length and the ability to synthesize "new" information in context, given the original piece of data and prior knowledge. To provide some evidence that SEAL might show greater benefits over *base-model* synthetic data, we've run an additional experiment scaling *down* to 3B. See our results given in the response to reviewer oBBv. The key point is that the the relative increase in performance with SEAL over using the base model for synthetic data generation increases with model size.
>
> 7. **Self-Edit Formats Predefined.** We've added results with no predefined self-edit format, as well as a variety of other prompting formats (see our rebuttal to reviewer nQa9). Predefined prompting formats won out in our experiments, but SEAL improved performance by similar margins *no matter the prompt (or lack thereof)*.
>
>
> **Response to Questions:**
>
> 1. **What Makes a Good Self-Edit.** We’ve noted that self-edits in knowledge-incorporation are longer. However, as we explain in the new results and response to reviewer nQa9, this does not account for all of the gains. Qualitatively, in the case of the “implications” prompt, good self-edits seem to contain a long but diverse list of short atomic statements that follow from the base text. We will also expand the appendix with additional qualitative examples and will provide a link to the repo containing all of the data for any future analysis.
>
> 2. **Proxy Reward.** Note that RL would not be done *throughout* pretraining. We see it more as a warmup such that the model can then self-edit effectively (or produce synthetic data for a student model). However, we think that using a proxy reward models is a good idea! As with any RL setup, we could train a reward model and then use that instead, or we could try to use a faster-to-compute proxy metric for what is a good self-edit. To explore this idea, we've run another experiment using a rubric-based reward, where we query gpt-4.1 to score the length, diversity, quality, and correctness (whether it follows from the contexts) of the response, and use this as a reward rather than our current inner-loop. This is significantly faster than our original inner loop, but performance did not reach the level of the full meta-learning inner loop.
>
> ## Additional Experiment: Proxy Reward
>
> We experiment with replacing the inner loop with a proxy reward based on a human-crafted and tuned rubric with 4 categories: length, diversity, quality, and correctness. A gpt-4.1 grader model ranked each category on a 1-5 scale, and the sum of these scores was used as the RL reward. The final results are shown below, as well as the overall RL training times.
>
> | Model                |  Base | Post-RL | Time   |
> | :-------------------- | ----: | ------: | :------ |
> | SEAL                 | 32.0% |   **47.0%** | ~6 hr  |
> | SEAL w/ Proxy-Reward | 32.0% |   45.6% | **~5 min** |
>
> We believe further rubric/metric tuning could lead to a stronger reward signal. However, the benefit of the full SEAL loop is that this is not necessary, and we directly learn what edits are useful for the base model. We believe both approaches are promising when scaling to larger model sizes and greater compute budgets.

---

> > ### Comment · Reviewer_hhPg · 2025-08-04
> >
> > I appreciate the effort the authors have put into addressing the concerns raised by myself and other reviewers. While the authors' responses and new results are valuable, they do not sufficiently alleviate my concerns about the work's practical significance and the robustness of its claims.
> >
> > My primary concern remains the immense computational cost and the reliance on a well-defined, immediate reward signal. The authors argue the cost is a fixed, one-time investment. While true, this investment is prohibitively high, potentially limiting the method to only the most well-resourced labs. More importantly, this reframing sidesteps the issue of applying SEAL to real-world, continuous adaptation scenarios where such a costly pre-training of the policy is not feasible. The new experiment with a proxy reward model is a good step, but it also highlights a critical trade-off: the closer the method gets to being practical (fast), the less effective it becomes (lower performance).
> >
> > The authors' rebuttal acknowledges that the learned policy is domain-specific (e.g., QA) and does not claim generalization to other tasks like code or math. This is a reasonable clarification, but it also confines the significance of the work. The paper is framed as a general framework for "Self-Adapting Language Models," but the evidence only supports its efficacy within a narrow, pre-defined problem structure.
> >
> > The new, larger-scale CPT experiments are useful, but they operate in a batch-update regime. This is fundamentally different from the sequential, continual learning scenario where catastrophic forgetting (which the authors admit is a problem) is the main challenge. Therefore, these results do not adequately demonstrate the method's utility for the very problem of ongoing adaptation that motivates the paper. The comparison with Generative Adapter (a hypernetwork approach) in their rebuttal is revealing: Generative Adapter substantially outperforms SEAL in the single-document (n=1) setting (66.8% vs 47.0%), which is the core test-time adaptation scenario. While SEAL performs better in the batch CPT setting, this suggests that for the task of rapid, single-instance adaptation, more direct methods may be superior. This complicates the narrative about SEAL's overall effectiveness.

---

> > > ### Author Response · Authors · 2025-08-04
> > >
> > > Thank you for the thoughtful follow-up and for pushing us on practicality and scope. We address your points and clarify how we will revise the framing in the paper.
> > >
> > > **Scope and positioning.**
> > > We agree that our experiments support SEAL most clearly in *integrating knowledge from documents* and *few-shot learning* under predefined edit formats. We will revise the wording of our introduction to avoid over-general framing. Our claim is that SEAL is a **framework for learning to generate and apply self-edits** that improves downstream performance in these settings—not that it already solves open-ended continual adaptation across domains like code or math.
> > >
> > > **On computational cost and reward design.**
> > > As compute continues to scale, we believe methods that enhance results per unit of *data*—even when currently constrained by compute—will become increasingly important.
> > >
> > > Although computationally demanding in its current form, our work introduces a promising and entirely novel direction of meta-learning approaches with synthetic data generation. We view this work as a foundational step showing that a model can learn to produce finetuning data that benefits itself, (whether performed individually or in batches) which could lay the foundation for future, more efficient methods.
> > >
> > > There are several potential extensions to this work that we believe are interesting that may address your longer-term doubts about the method. Two of these directions are: using a reward based on the adapted model's perplexity on related data (which no longer requires having documents paired with questions), and using a reward based on proxy metrics (which greatly reduces computational cost). In particular, our results with proxy metrics were not heavily tuned and could possibly even be improved over using the "true" method of rewarding generations that directly improve the model after a finetuning step. Overall, the potential for these methods is high, and we do not believe the current computational cost is likely to be a long-term barrier.
> > >
> > > **Batch vs. sequential continual learning.**
> > > You are right that our strongest results are in the *batch* regime and that we do not yet address the core mechanisms needed for streaming updates.
> > >
> > > To clarify, sequential continual adaptation was *not* the main motivation for this paper, but rather an exploratory extension. While we included experiments on sequential continual learning in the Limitations section to preliminarily investigate SEAL's performance in such settings, our core method does not explicitly target streaming updates (e.g., optimizer-state carry-over, replay, etc.) anywhere. We will make this more clear by rephrasing the two paragraphs in the section on catastrophic forgetting.
> > >
> > > **Comparison to Generative Adapter (GA).**
> > > We believe the hypernetwork approach differs fundamentally from our self-editing methodology, and both are worth investigating. A higher performance with hypernetworks on single-instance adaptation tasks doesn't alter our narrative. For instance, storing documents and doing RAG would also achieve very high scores on this task, yet their underlying mechanisms and motivations are entirely different. Our focus remains firmly on models that meta-learn how to create training data for themselves with broad applicability, rather than optimizing solely for single-instance adaptation. The benefit of being able to adapt in response to many instances is quite significant, since this is the more practical case (if we had only a single document, we could've just put it in context).
> > >
> > > Finally, note that with our additional prompting experiments, we improve from 47.0% to 55.6%. We believe our method has great potential for growth and that the high single-document score with GA does not hinder this.

---

> > > > ### Comment · Reviewer_hhPg · 2025-08-06
> > > >
> > > > Thanks for your response. It clarified most of my concerns. I believe that more efficient and effective streaming updates remain a challenge. If LLMs can learn something after each conversation and automatically trigger batched/streaming updates, they can scale better. I'd also like to know how SEAL compares to current memory-based approaches.

---

> > > > > ### Author Response · Authors · 2025-08-06
> > > > >
> > > > > We are glad that our response clarified most of your concerns!
> > > > >
> > > > > We agree that streaming updates will be an important challenge for future work. There are many different approaches for doing persistent memory updates, but we are most interested in weight adaptation. We hope our work is a valuable step towards this paradigm.

---

### Official Review · Reviewer_D8Zo · 2025-07-02

**Clarity:** 3
**Significance:** 4
**Originality:** 3
**Rating:** 5
**Confidence:** 3

**Summary:**

This paper studies the direction of enables LLMs to self-adapt by automatically editing the data format. First, the model generates the refined context and then adapts the original LLM on this context. The contexts, which correspond to improved downstream performance, are marked as high rewards, and the model would be updated to maxmize these rewards.

**Questions:**

Q1. In Step 4 of Algorithm 1, how can the authors ensure that the self-edit generated based only on $C$ is suitable for downstream $Q$? Is it possible that one context $C$ corresponds to multiple different $Q$, making a context-only self-edit suboptimal for all associated $Q$? Would it be more appropriate to generate self-edits conditioned on both $C$ and $Q$? Furthermore, how is the type of self-edit (e.g., rewriting, augmentation) selected or controlled, given that different $Q$ types may require different editing strategies?
Q2. The formula is missing a sequence number.
Q3. In Section 4.1 and 4.2 (the main results), the adopted baselines are too simple. None of the recent work in Related work are used in this experiment, making the results more like ablation study.
Q4. In few shot learning, what will happen if the self-edit does not include optimization configurations, e.g., learning rate, number of training epochs?
Q5. The authors have give some examples to intuitively explain why editing the data format is helpful. However, an in-depth theoretical analysis can be more convincing and improve the quality of this paper.

**Ethical Concerns:**

["NO or VERY MINOR ethics concerns only"]

**Final Justification:**

I recommend Accept. My main concerns are addressed after author rebuttal.

**Limitations:**

yes

**Quality:**

3

**Strengths And Weaknesses:**

Strengths:
S1. The studied problem is important and interesting.
S2. The paper is well-motivated and easy to follow.
S3. The proposed method is novel.

Weaknesses:
W1. The methodology lacks clarity on how context-specific (context denoted as $C$) self-edits generalize across multiple downstream tasks (downstream tasks denoted as $Q$).
W2. The formula is missing a sequence number.
W3. Lack of competitive and recent baseline methods.
W4. The contribution of different components of self-edits (e.g., optimization configurations) is unclear.
W5. Lack of in-depth theoretical analysis about why editing the data format can benefit the downstream performance.

The details and corresponding questions about these weaknesses can be found in Questions below.

---

> ### Author Rebuttal · Authors · 2025-07-31
>
> We thank the reviewer for their constructive feedback! We address the reviewers weaknesses and questions below.
>
> **Response to Weaknesses/Questions:**
>
> 1. **Prompting With Both Context and Questions.** Prompting the model on both C (context) and Q (downstream task) might be useful in some settings, but we wish to be able to self-edit in response to any document C without knowing ahead of time what downstream tasks Q we will encounter. We also agree that our current setup uses task-specific self-edit strategies, while further work might allow for the model to employ an even larger pool of strategies in response to different types of data. One potential extension, as you suggest, is to train across a diverse set of tasks, each with its bespoke self-editing method, and allow the model to learn to select an appropriate self-editing strategy for new tasks.
>
> 2. **Typo in Formula.** Could you clarify which formula you are referring to?
>
> 3. **Comparison with Related Work.** Thank you for the feedback. We’ve added comparisons to a number of recent baselines and prompting methods for generating synthetic data for a task. See the table in our response to reviewer nQa9 for these new results. The key point is that some prompting methods do better than "implications," but RL on top of these prompts improves accuracy further, by similar margins. Entigraph, a human-crafted synthetic data generation pipeline, does comparably to SEAL in our continued pretraining setup. Generative-Adapter, which aims to directly generate the LoRA adapter weights, does better in the single-document regime, but has many other limitations. See our discussion at the bottom of our response to reviewer nQa9.
>
> 4. **Contribution of Components of Self-Edits.** Without training for self-edit generation, performance in the few-shot domain drops to 20%. For each of the parameters such as learning rate, number of epochs, etc., setting them far off the “sweet-spot” leads to a substantial decrease in performance. In this sense, all of these parameters are important. It’s unclear what would be a “fair” point to fix one while letting the model generate the other, or what this would tell us, so we did not run any additional experiments here. These experiments are meant to show that signal from transiently adapted models can reinforce better self-generated hyperparameter selection through RL, which we believe was shown.
>
> 5. **Theoretical Analysis.** While we do not currently provide theoretical contributions, we agree that this represents an interesting direction. Prior work ([1], Yang et al., Synthetic Continued Pretraining) demonstrates from a theoretical perspective how strategically rewriting data can greatly improve the scaling behavior of learning on a set of data. Here, we do not enforce any particular structure such as in [1], so it’s hard to give additional theoretical insight. Instead, we see our method as towards automating the discovery of good synthetic data formats such as that identified in [1].

---

> > ### Comment · Reviewer_D8Zo · 2025-08-03
> >
> > Thanks the authors for the responses. The response to W1 should be clear in the final paper. Additionally, numbering the equations would improve readability and make it easier to reference them (for W2).

---

> > > ### Author Response · Authors · 2025-08-03
> > >
> > > Thank you for the suggestions. We will make sure our response to W1 is in the final paper along with equation numbers.

---

### Official Review · Reviewer_nQa9 · 2025-07-03

**Clarity:** 4
**Significance:** 3
**Originality:** 4
**Rating:** 5
**Confidence:** 4

**Summary:**

The paper proposes a new framework, SEAL, for model learning from self-generated data. SEAL includes a RL phase to optimize data generation, aiming to better adapt the model to given downstream tasks. The authors analyze the effectiveness of SEAL by evaluating it on two tasks: the few-shot ARC benchmark and knowledge incorporation.

**Questions:**

I am curious whether model collapse becomes a problem when training on self-generated data. Is this related to the number of adaptation steps or the amount of synthetic data used during adaptation?

**Ethical Concerns:**

["NO or VERY MINOR ethics concerns only"]

**Final Justification:**

How to generate synthetic data for a given task is an important question in practice. The paper provides an RL framework to improve data synthesis, given the downstream evaluation as the reward. The paper has a bit of weakness in its evaluation setup. They only evaluate on two tasks: knowledge injection and few-shot learning, while in practice, we can imagine there are many more scenarios that involve adaptation, like personalization chatbots and customized tool agents where the model needs to learn user preferences, procedural knowledge, and new function calls, etc. Also, they train a data generator for each task separately. This limits their method from being used as a general tool to adapt the model to any task. Despite this, during the rebuttal, the authors provide a thoughtful discussion on a comparison with human-designed data synthetic pipelines and other model adaptation paradigms like hypernetworks, which significantly strengthens the paper. So, I recommend a score of 5.

**Limitations:**

yes

**Quality:**

3

**Strengths And Weaknesses:**

Strength

- Self-adaptation is an essential ability in human learning, but effective methods for enabling it in transformer-based language models are currently lacking.
- Learning from self-generated data is a general and practical direction for enabling self-adaptation.

weakness

- Although self-adaptation is an important challenge, SEAL only explores a narrow case: test-time adaptation to new instances of the same task seen during RL training, rather than to unseen tasks. This limits the broader significance of the paper. Both training and testing focus on visual reasoning and knowledge incorporation. It would be more compelling to include experiments where RL is applied on few-shot tasks from benchmarks A, B, and C, and then tested on distinct benchmarks X, Y, and Z.
- Given that the experiments focus on adapting to new instances within known tasks, it is unclear how much the RL training improves over carefully designed synthetic data pipelines crafted by humans. For example, in knowledge injection, [1] seems to provide a more structured approach for document augmentation. In few-shot learning, it would be interesting to see comparisons with synthetic data generated by different models.
- Some additional ablation studies would strengthen the paper. For instance, why was ReST chosen as the RL algorithm? Does the size of the RL training dataset affect performance? Is it necessary to generate different augmentation strategies or optimization configurations?
- Regarding parameter adaptation from demonstrations or instructions, prior work on hypernetworks should be discussed more thoroughly. For example, [2] is relevant and should be considered.

[1] Yang et al., "Synthetic Continued Pretraining." ICLR 2025

[2] Chen et al., "Generative Adapter: Contextualizing Language Models in Parameters with A Single Forward Pass." ICLR 2025

---

> ### Author Rebuttal · Authors · 2025-07-31
>
> We thank the reviewer for their attention and helpful feedback! We will address each weakness and question.
>
> **Response to weaknesses:**
>
> 1. **Generalization.** Our evaluation still uses an unseen test set from the distribution, and our model’s ability to self-edit generalizes within that scope. The general knowledge experiments were meant to simulate learning from arbitrary training data. We focused on document QA for its ease of verification, but concede that RL training on only this likely would not generalize to many other types of LM training data. We also don’t claim that self-editing on ARC tasks would transfer to entirely different few-shot learning tasks. However, we believe that even within these datasets, generalization is a challenge.
>
> 2. **Prompting & Heuristic Pipelines.** This is a great point. We have experimented with a variety of prompts and structured methods for synthetic data generation—ranging from different prompting strategies for SEAL or GPT‑4.1 to using entigraph [1] for continued pretraining—within the general knowledge incorporation domain. The table below includes the additional experimental results, showing that SEAL consistently delivers performance gains across each base prompting strategy, and matches or exceeds [1] for continued pretraining in our setting. This table will be included in the updated version of the paper.
>
> 3. **Algorithm Ablations.** We chose ReST-EM because it was simple and worked well, and we had a difficult time stabilizing GRPO in our meta-learning setting. However, to scale up SEAL, we agree that moving to more sophisticated RL methods would be beneficial. In the few-shot domain, we already know that augmentation strategies and the optimization parameters make a difference (e.g. moving any optimization parameters off of the sweet spot would lead to worse performance). It’s unclear what would be a “fair” point to fix one while letting the model generate the other, so we did not run any additional experiments here.
>
> 4. **Hypernetworks.** Thank you for pointing out the related work. We will include additional discussion on hypernetworks. We ran Generative Adapter [2] with our evaluation setup and have discussed the results at the bottom of this response.
>
> **Response to Questions:**
> 1. **Model Collapse.** Our experiment in the Limitations section shows that continual gradient steps (without maintaining optimizer states or batching) leads to model collapse. However, batching the updates recovers much of the performance, at least on the task at hand (see table 2 on continual pretraining experiments). Additionally, we’ve run larger continual pretraining experiments with full finetuning rather than LoRA and provided them in the table below. Notably, accuracy does not degrade but rather increases to 58.2% for n=200 and remains high at 46.4% for n=2067 when updates are batched, compared to training and evaluating a separate LoRA adapter for each document (47.0%).
>
> ## Main Additional Experiments
>
> Since there is no option for a global response, we provide the bulk of our additional results and explanations below, and reference them throughout our individual responses.
>
> ### Prompting
>
> | Method                     | Original | Round 1 | **Round 2** | gpt-4.1 synthetic |
> | -------------------------- | -------: | ------: | ----------: | ----------------: |
> | No self-edit               |     33.5 |       – |           – |                 – |
> | Implications               |     39.7 |    43.7 |    **47.0** |              46.3 |
> | Implications-long          |     49.3 |    52.4 |    **54.4** |              54.1 |
> | Implications-very-long     |     45.0 |    51.5 |    **52.1** |              40.9 |
> | Rewrite                    |     49.4 |    55.3 |    **55.6** |              54.4 |
> | Self-QA                    |     37.3 |    42.8 |    **48.7** |              39.2 |
> | No Prompt                  |     13.8 |    12.7 |    **18.9** |              28.6 |
> | Chain-of-thought-self-edit |     38.7 |       – |           – |                 – |
> | Chain-of-thought-eval      |     37.8 |       – |           – |                 – |
>
> A central critique was that our self-adaptation procedure is not sufficiently general, or that we did not compare with other methods for generating synthetic data. In the knowledge QA task, we’ve added results showing that SEAL remains effective across a variety of self-editing instructions. This demonstrates that within the general knowledge incorporation domain, SEAL is not tied to a single self-edit format.
>
> The “implications-long” and “implications-very-long” columns prompt for longer generations.  While none of the reviewers pointed this out explicitly, we wanted to test whether prompting could match the gains from RL, since we saw that RL training naturally increased the lengths of generations (for example, see Figure 5). We find that many prompts achieve higher scores than our main “implications” prompt, but that RL on top of them increased results by similar margins, meaning that the improvement could not simply be attributed to length.
>
> Here, “chain-of-thought-self-edit” refers to having the model reason before providing the synthetic data, while “chain-of-thought-eval” refers to having the model reason before answering the questions (letting the model “pull out” information from its weights). Both are done with prompting for "implications." The "gpt-4.1 synthetic" column refers to finetuning the base Qwen2.5-7B model using data generated by gpt-4.1 using the corresponding prompt.
>
> Further analysis is given in specific responses to reviewers.
>
> ### Synthetic Continued Pretraining
>
> | Method                    | Continued Pretraining (n=200) | Continued Pretraining (n=2067) |
> | ------------------------- | ----------------------------: | -----------------------------: |
> | Base                      |                         32.0 |                          29.0 |
> | Train on Passage            |                         36.0 |                          31.2 |
> | Train on Passage + Base Model Synthetic Data           |                         50.6 |                          43.4 |
> | SEAL                      |                     *58.2* |                          *46.4* |
> | Entigraph (pairs)         |                         46.2 |                          38.6 |
> | Entigraph (pairs+triples) |                         56.0 |                          *48.6* |
> | Train on Passage + GPT-4.1 Synthetic Data         |                     **59.4** |                      **49.2** |
>
> Here, we run additional continued pretraining experiments. These experiments are done with the same methodology as the ones in the original paper, except we do full finetuning rather than LoRA. We experiment with n=2067 documents, which is the full validation set. Entigraph [1] has two settings—the second (pairs+triples) creates 10 generations per document, while the first (pairs) and all other methods create 5.
>
> Surprisingly to us, with full finetuning, we find that performance *improved* over the n=1 regime in which we RL-trained our model to self-edit. Performance was highest using a batch of n=200 documents, but remained strong at n=2067. We conjecture that this is due to issues with gradient updates over small amounts of data that are encountered with n=1 document, even with strong hyperparameters.
>
> ### Comparison to Generative Adapter
>
> | Model              | Base | TTT (n=1) | CPT (n=200) |
> | ------------------ | ---: | --------: | ----------: |
> | SEAL               | 32.0 |      47.0 |    **58.2** |
> | Generative Adapter | 24.4 |  **66.8** |        28.0 |
>
> We ran Generative Adapter [2], a hypernetwork approach to generating a LoRA adapter based on context, with our evaluation setup. All 6 values in the table are on the same evaluation set, but CPT batches updates over all documents while TTT trains an adapter and evaluates each one separately. We used the Mistral-7B-based model for Generative-Adapter, and it achieved a performance of 66.8% on n=1 document, outperforming SEAL, and 28.0% on n=200 continued pretraining, greatly underperforming SEAL.
>
> However, we would like to highlight some key differences between the methods. SEAL parameterizes weight updates in the generation of training data, which (1) allows for reusing data for continued pretraining or on arbitrary models and (2) allows models to leverage reasoning as documents scale in size and complexity. SEAL is also not restricted to LoRA finetuning, and is much more general in that it potentially allows for many different types of updates given a base document beyond just memorizing a piece of data. For example, perhaps we wish to learn and do a weight-update from an environment/user interaction. It is unclear how hypernetworks could scale to such tasks, while generating data for next-token prediction could leverage the in-context learning abilities of the model.

---

> > ### Author Response · Authors · 2025-08-06
> >
> > We thank the reviewer again for their feedback. We believe we addressed many of the concerns, and would greatly appreciate if the reviewer could look over our response and let us know if they think there are any further areas for improvement.

---

> ### Comment · Reviewer_nQa9 · 2025-08-06
>
> Thanks for your clear response. You have addressed all my questions, and I will increase my assessment to 5.

---

### Decision · Program_Chairs · 2025-09-17

**Decision:**

Accept (poster)

**Comment:**

Overview:

This paper proposes SEAL, a method for models to learn from self-generated data.  The proposed method incorporates model-generated self-edits, followed by evaluation on a target task, and a subequent model update based on target task performance.  Effectiveness of SEAL is demonstrated on two downstream tasks and outperforms training on other synthetic datasets such as that of GPT-4.


Strengths:

- The paper is very well-written.  The Figures 1-3 add a good amount of clarity to the method.  The method itself is also intuitive to understand.

Weakness:

- Experiments are run mostly at smaller scales including 3B and 7B model sizes, and on smaller scale evaluations.  The experiments are also only limited to two tasks which are somewhat non-standard compared to other evaluations at this size such as Alpaca, MT-Bench, IF Eval, MMLU, etc.

- Reviewers pointed out that there are some missing comparison with more SOTA work. As the paper primarily focuses on learning from self-generated data, the best comparison at the time was likely GPT-4 generated data as an upper bound on synthetic data generated for improvement.  While there were still some concerns about the comparison of the GPT generated data, the performance comparisons are reasonable and encouraging.

Main Reasons to Accept or Reject:

- Accept: A clear, well-written paper with a simple and intuitive method that performs well.  Authors have added several experiments expanding comparison.

- Reject: Experiments are only limited to two tasks, which are somewhat non-standard and limited in size (initially 200 but expanded in rebuttal).  The method is general but not applied in more settings.  The authors note it could be applied in math, code, etc. but do not give any particulars.

Rebuttal Period:

All reviewers engaged in the rebuttal leading to increases scores where all reviewers are positive.  Reviewers note that most of their concerns have been addressed by the rebuttal and the authors have run quite a few additional experiments with another model and more baselines, and addressed some potential limitations such as run time.  However, the largest outstanding concern to me is the lack of multiple evaluation sets.  This seems like a general method that could be applied in more domains and on any generation benchmark as self-generating data for a target task has general use.

Final Recommendation:

The paper proposes a novel method for training language models on self-generated data. There are a few comments from reviewers and additional experiments that the authors should incorporate into future versions.